# Impact of Elicitation on Plant Antioxidants Production in *Taxus* Cell Cultures

**DOI:** 10.3390/antiox12040887

**Published:** 2023-04-05

**Authors:** Edgar Perez-Matas, Pascual Garcia-Perez, Mercedes Bonfill, Luigi Lucini, Diego Hidalgo-Martinez, Javier Palazon

**Affiliations:** 1Department of Biology, Healthcare and the Environment, Faculty of Pharmacy and Food Sciences, University of Barcelona, 08028 Barcelona, Spain; 2Department for Sustainable Food Process, Università Cattolica Del Sacro Cuore, Via Emilia Parmense 84, 29122 Piacenza, Italy; 3Nutrition and Bromatology Group, Department of Analytical and Food Chemistry, Faculty of Food Science and Technology, Ourense Campus, Universidade de Vigo, 32004 Ourense, Spain

**Keywords:** plant antioxidants, phenolic compounds, *Taxus* cell cultures, elicitation, metabolic fingerprint

## Abstract

Elicited cell cultures of *Taxus* spp. are successfully used as sustainable biotechnological production systems of the anticancer drug paclitaxel, but the effect of the induced metabolomic changes on the synthesis of other bioactive compounds by elicitation has been scarcely studied. In this work, a powerful combinatorial approach based on elicitation and untargeted metabolomics was applied to unravel and characterize the effects of the elicitors 1 µM of coronatine (COR) or 150 µM of salicylic acid (SA) on phenolic biosynthesis in *Taxus baccata* cell suspensions. Differential effects on cell growth and the phenylpropanoid biosynthetic pathway were observed. Untargeted metabolomics analysis revealed a total of 83 phenolic compounds, mainly flavonoids, phenolic acids, lignans, and stilbenes. The application of multivariate statistics identified the metabolite markers attributed to elicitation over time: up to 34 compounds at 8 days, 41 for 16 days, and 36 after 24 days of culture. The most notable metabolic changes in phenolic metabolism occurred after 8 days of COR and 16 days of SA elicitation. Besides demonstrating the significant and differential impact of elicitation treatments on the metabolic fingerprint of *T. baccata* cell suspensions, the results indicate that *Taxus* ssp. biofactories may potentially supply not only taxanes but also valuable phenolic antioxidants, in an efficient optimization of resources.

## 1. Introduction

An important goal of the United Nations 2030 Agenda for Sustainable Development is “to achieve the sustainable management and efficient use of natural resources” (goal 12.2) “https://www.un.org/sustainabledevelopment/sustainable-development-goals/ (accessed on 1 March 2023)”. Sustainability includes the use of plant raw materials with minimal generation of waste, which in a circular economy is converted into an added-value resource “https://www.europarl.europa.eu/news/en/headlines/economy/20151201STO05603/circular-economy-definition-importance-and-benefits (accessed on 1 March 2023)”. In this pipeline, biofactories based on plant cell and organ cultures constitute a renewable source of natural products with antioxidant, anticancer, and other beneficial biological activities [1].

*Taxus* spp. cell cultures represent an efficient biosustainable system for the production of taxanes, most notably paclitaxel, which are used as therapeutic anticancer agents. However, many other specialized metabolites have been isolated and identified in *Taxus* plants [2,3,4] and may also be produced by *Taxus* cell cultures. Among them, phenolic compounds such as flavonoids, phenolic acids, tannins, lignans, and coumarins are the most important, due to their potent antioxidant capacity [5,6,7,8]. As consumer habits change, there is a growing market demand for these natural antioxidants, valued as substitutes for synthetic additives with toxic and carcinogenic side effects, as well as for their wide range of biological activities [9]. In the context of environmental sustainability, phenolic compounds have been extracted from plant raw materials, as well as agricultural and processing by-products [10].

Phenolic biosynthesis in plants is governed by the phenylpropanoid pathway, where the conversion of the amino acid phenylalanine into *trans*-cinnamic acid by the enzyme phenylalanine ammonia-lyase (PAL) marks a shift in carbon flux from primary to secondary phenylpropanoid metabolism [11]. As antioxidants are usually produced in low concentrations in plants, elicitation of cell cultures is a widely used strategy that efficiently enhances specialized metabolite production through the stimulation of plant stress responses [12]. Additionally, in connection with the 2030 Agenda for Sustainable Development, the large-scale production of bioactive compounds in plant cell-based biotechnological platforms is an efficient and eco-friendly process, due to rapid cell growth, controlled growing conditions, and easy scale-up [13]. Among the wide range of elicitors available, salicylic acid (SA) and the jasmonic acid analogue coronatine (COR), two compounds naturally involved in the regulation of plant growth and stress tolerance, have been extensively utilized for this purpose [14]. 

To date, only a few studies have explored *Taxus* cultures as a source of specialized metabolites other than taxanes. Thus, the biosynthetic potential of two elicited *Taxus × media* hairy root lines for the in vitro production of six lignans of therapeutic interest was studied [15]. Their results demonstrate that *Taxus* spp. can be successfully used as a biotechnological platform to obtain valuable phenolic compounds, although more research is needed on the effects of elicitors on their biosynthesis and how the phenolic profile and content change over time. In this context, the application of untargeted metabolomics enables a systematic analysis and comparison of primary and specialized metabolism, and sheds light on the metabolic variations induced by elicitation [16]. 

In this work, a powerful combinatorial approach based on elicitation and untargeted metabolomics was applied to study the effects of the elicitors COR (1 µM) and SA (150 µM) on the metabolic fingerprint of *Taxus baccata* cell suspensions, based on their phenolic profile. With the aim of efficiently optimizing resources, we explored the potential of *Taxus* plant cell biofactories as a new biosustainable source of polyphenols with antioxidant activity, besides producing anticancer compounds.

## 2. Material and Methods

### 2.1. Plant Material and Establishment of Cell Suspensions

Cell suspension cultures were established from a stable friable callus line of *T. baccata*, as previously described in [17]. After surface sterilization, a callus line was obtained by placing the explants in contact with the callus induction medium, which consisted of Gamborg’s B5 medium (B5) [18] supplemented with 2x B5 vitamins, 3% (*w/v*) sucrose, and the following growth regulators: 4 mg/L 2,4-dichlorophenoxyacetic acid, 1 mg/L kinetin, and 0.5 mg/L gibberellic acid. When callus tissues started to form after 3–4 weeks, they were separated from the explants and placed together for further growth in solid B5 supplemented with 2x B5 vitamins, 0.5% (*w/v*) sucrose, 0.5% (*w/v*) fructose, and the following growth regulators: 2 mg/L 1-naphthaleneacetic acid, 0.1 mg/L 6-benzylaminopurine, and 0.5 mg/L gibberellic acid. Prior to autoclaving, the pH of all the media was adjusted to 5.8. After autoclaving, an antioxidant solution [19] composed of 14.6 g/L L-glutamine, 2.5 g/L ascorbic acid, and 2.5 g/L citric acid was added to all media. 

Calli were then kept at 25 °C in darkness and subcultured every two weeks under the same conditions. To establish *T. baccata* cell suspensions, a two-stage system was used as described in [17]. Briefly, after 14 days of culturing the cells in liquid growth medium, 3 g of cells was transferred to 10 mL of taxane production medium in a 175 mL flask capped with a MagentaTM B-cap (Sigma Aldrich, St Louis, MO, USA). This medium consisted of B5 liquid medium supplemented with 3% (*w/v*) sucrose and the following growth regulators: 2 mg/L picloram, 0.1 mg/L kinetin, and 0.5 mg/L gibberellic acid at pH 5.8. Cultures were then maintained in darkness at 25 °C and 100 ± 1 rpm in an orbital shaker incubator (Adolf Kühner, Birsfelden, Switzerland).

### 2.2. Elicitation Experiments and Biomass Determination

Cell suspension cultures were elicited at the beginning of the second phase of culture [20] with 1 µM of coronatine (COR) (Sigma Aldrich, St Louis, MO, USA) or 150 µM of salicylic acid (SA) (Sigma Aldrich, St Louis, MO, USA), which were previously filter-sterilized. Cultures were incubated in the same conditions as described above, and samples were harvested at 8, 16, and 24 days of treatment. Fresh weight (FW) was determined by filtering the cells with 80 µm nylon filters, and cells were immediately frozen at −20 °C. The cells were then lyophilized and powdered to determine dry weight (DW).

### 2.3. Sample Extraction for Untargeted Metabolomics 

Lyophilized cell samples were powdered and subjected to solvent extraction using a mixture of MeOH/H_2_O/HCOOH (80:19.9:0.1 *v/v/v*) at a concentration of 50 mg/mL, vortexed for 2 min until homogenization, and then placed in an ultrasonic bath for 30 min at 25 °C. After centrifugation at 12,000× *g* for 10 min at 4 °C (Eppendorf 5810R, Hamburg, Germany), the supernatants were collected and syringe-filtered (0.22 µm PVDF filters, Millipore, Billerica, MA, USA). Finally, the extracts were transferred into vials and stored at −20 °C until analysis. 

### 2.4. Phenolic Profiling via UHPLC/QTOF-MS Approach 

The phenolic profile of *T. baccata* cell samples was determined by an untargeted metabolomics approach, which was performed using a tandem system consisting of a JetStream electrospray ionization source and a quadrupole time-of-flight mass spectrometer (G6550 QTOF) coupled to a 1290 ultrahigh-performance liquid chromatography system (UHPLC/QTOF–MS; Agilent technologies, Santa Clara, CA, USA). Firstly, the chromatographic separation was carried out using a reverse-phase Agilent ZORBAX Eclipse Plus C18 column (2.1 × 100 mm, 1.8 µm). Elution was achieved by applying a continuous linear gradient, employing a binary mobile phase system that consisted of a mixture of acetonitrile (solvent A) and water (solvent B) for 32 min, in the following profile: 0–32 min, form 6%A, 94%B to 94%A, 6%B. An equilibration period of 3 min was employed to return to the initial elution condition prior to the subsequent injection. Flow rate was set at 200 µL/min. The injection volume was 12 µL, and each sample analysis was repeated twice (n = 6). For compound detection, the mass spectrometer operated in positive polarity and SCAN mode, and the extended dynamic range mode was set at 100–1200 *m/z* and a nominal resolution of 40,000 FWHM [14]. 

The annotation of phenolic compounds was achieved according to their isotopic pattern (including monoisotopic mass, isotopic spacing, and isotopic ratio) and expressed as means of the overall identification score using the “find-by-formula” algorithm by MassHunter Profiler v. 10.0 software (Agilent technologies, Santa Clara, CA, USA) and the database Phenol-Explorer 3.6. Finally, compound annotation was carried out according to Level 2 (putatively annotated compounds) as displayed by the COSMOS Metabolomics Standards Initiative [21]. Once annotated, the compounds were filtered in relation to their abundance, only selecting those detected in 66.6% of the replicates within each treatment. The detected phenolic compounds were grouped into different families and subfamilies and quantified through calibration curves obtained with reference standards for each class, as previously indicated [14]. Tyrosol was selected as the reference standard of low-molecular-weight phenols (LMW), including alkylphenols, coumarins, phenylpropenes, quinones, tyrosols, and other phenolics. Although some flavonoids and phenolic acids can be classified as LMW compounds, we have excluded them from this group in our present work. This is because these compounds are best represented and quantified using reference standards other than tyrosol. Ferulic acid was selected for phenolic acids; sesamin for lignans; *trans*-resveratrol for stilbenes; and luteolin for flavonoids, including flavones, flavonols, anthocyanins, flavanols, flavanones, isoflavonoids, chalcones, and dihydrochalcones. The results of the semi-quantification were expressed as the equivalents of each reference standard in µg/g of dry weight (DW), as the mean ± standard deviation (n = 6). 

### 2.5. Determination of Antioxidant Activity by DPPH Radical Scavenging Assay

For the antioxidant activity determination of the *T. baccata* cell cultures, 10 mg of the lyophilized samples was dissolved in 1 mL of MeOH/H_2_O (80:20 *v*/*v*), then vortexed and sonicated for 10 min. Extracts were then centrifuged for 20 min at 14,000× *g* (Eppendorf 5810R, Hamburg, Germany), and the supernatants were transferred into vials, diluted 1:100, and stored at −20 °C until use. 

The 2,2′-Diphenyl-1-picrylhydrazyl (DPPH) assay and half-maximal effective concentration (EC_50_) determinations were performed according to the literature [22]. Briefly, 1 mL of a fresh working solution of 0.208 mM DPPH in methanol was added with 800 µL of Tris-HCl buffer 0.1 M (pH 7.4) in a testing tube. Then, 200 µL of each testing extract was added and mixed quickly. The solution was kept at room temperature for 30 min. The absorbance of the mixture was measured spectrophotometrically at 517 nm. Additionally, a Trolox (6-hydroxy-2,5,7,8-tetramethylchroman-2-carboxylic acid) calibration curve with concentrations ranging from 0 to 300 µM was used as a positive control. The calculation of % scavenging of DPPH· radical was as follows:(%) DPPH radical scavenging activity = [(A1 − A2)/A1] × 100

The A1 is the absorbance of DPPH· radical + MeOH, and A2 is the absorbance of testing sample solution. The EC_50_ values were obtained using the regression line of each set of samples at different concentrations according to the literature [22]. 

### 2.6. Bioinformatic Analysis and Statistics

Raw data obtained from metabolomic profiling were statistically analyzed and interpreted by the software tool Mass Profiler Professional v. 12.6 (Agilent technologies, Santa Clara, CA, USA) as previously described [14]. Briefly, the abundance of each compound was transformed into log2 values and normalized at the 75th percentile. Then, multivariate unsupervised hierarchical cluster analysis (HCA) was carried out (Euclidean distance, Ward’s linkage rule) by the construction of a fold-change-based heat map, establishing a baseline against the median of all samples. Afterwards, a fold change (FC) analysis was carried out to reveal featured variations in the abundance of significant compounds, setting the threshold FC = 2. Later, supervised modeling by orthogonal projection to latent structures discriminant analysis (OPLS-DA) was performed by SIMCA v. 16.0.2 software (Umetrics, Sweden) to discriminate the effect of different elicitors over time. The OPLS models were combined with variable importance in projection (VIP) analysis to identify the markers responsible for the discrimination between treatments over time, the so-called VIP markers (VIP score threshold = 1.0). The quality of OPLS models was evaluated in terms of goodness-of-fit and goodness-of-prediction parameters (R^2^Y and Q^2^, respectively), and further validated statistically through cross-validation analysis of variance (CV-ANOVA, α = 0.05), and overfitting was excluded by the corresponding permutation tests. For comparative means, VIP markers were also analyzed by Venn diagrams using the online tool “jvenn” (available at http://jvenn.toulouse.inra.fr/app/index.html, accessed on 1 March 2023).

### 2.7. Statistical Analysis 

Results concerning cell growth and antioxidant activity were expressed as the average of three independent determinations ± standard deviation (n = 3). The statistical analysis was performed using Excel and RStudio software, and data were analyzed by multifactorial ANOVA followed by Tukey’s post hoc multiple comparison tests. The results for phenolic compounds semi-quantification were analyzed by one-way ANOVA, followed by Duncan’s post hoc test using SPSS 25 software version 25. In all cases, significant differences were assumed at *p*-value <0.05.

## 3. Results 

### 3.1. Untargeted Metabolomics of Elicited T. baccata Suspension Cell Cultures and Unsupervised Multivariate Analysis 

The untargeted phenolic profile of elicited *T. baccata* cell suspension cultures was performed by UHPLC/QTOF-MS to decipher the effects of two elicitors (1 µM COR and 150 µM SA) over time. The annotation list revealed 83 phenolic compounds, which were grouped into the following families: 28 flavonoids, 16 phenolic acids, 8 lignans, 4 stilbenes, and 27 other polyphenols. Regarding subfamilies, the flavonoid family included two anthocyanins, two chalcones, two dihydrochalcones, nine flavanols, four flavanones, five flavones, two isoflavonoids, and two dihydroflavonols. The class defined as other polyphenols consisted of five tyrosols, five hydroxycoumarins, four phenolic terpenes, three hydroxybenzaldehydes, two other polyphenols, one alkylphenol, one furanocoumarin, three hydroxyphenylpropenes, two alkylmethoxyphenols, and one hydroxycinnamaldehyde. The full list of annotated compounds, including retention time (min), mass (u), and molecular formula of each metabolite, is provided as Appendix A (Appendix A).

To provide insight into the effect of elicitation conditions on the phenolic profile of *T. baccata* cells, an unsupervised hierarchical cluster analysis (HCA) was performed, which revealed the phenolic metabolome-wide similarities and/or differences induced by each elicitor over time (Figure 1). In general, results indicate that both factors (elicitor and time) have a significant effect on the phenolic profile of *T. baccata* suspension cells. Regarding elicitor treatments, SA after 8 days induced a clear upregulation in the accumulation of a great number of compounds, mainly those belonging to flavonoids, phenolic acids (4-hydroxybenzoic acid 4-*O*-glucoside), and other polyphenols (carvacrol). Hence, this treatment clustered apart from the rest of the SA-treated time points, which showed a general down-accumulation of compounds, especially after 24 days of treatment. In contrast, COR treatments clustered together, suggesting a stronger effect over time of this elicitor on the phenolic profile. COR after 8 days exhibited the highest upregulation of compounds, principally flavonoids, such as 3-hydroxyphloretin 2′-*O*-glucoside or dihydromyricetin 3-*O*-rhamnoside, and lignans such as 7-hydroxymatairesinol or anhydro-secoisolariciresinol, but also some phenolic acids and other polyphenols. Interestingly, COR after 24 days showed a higher upregulation of compounds, including stilbenes (*d*-viniferin, pterostilbene, and piceatannol), phenolic acids (caffeic acid 4-*O*-glucoside and *p*-coumaroyltyrosine), flavonoids (6″-*O*-malonyldaidzin), and other polyphenols (esculin) than this elicitor at time point 16 days, which only clearly stimulated the biosynthesis of phenolic acids (*p*-coumaroyltartaric acid and 3-*p*-coumaroylquinic acid). Notably, control conditions at 8 days clustered together with SA at the same time point and separately from the other two time points (16 and 24 days), suggesting an influence of time of culture on the modulation of phenolic biosynthesis.

### 3.2. Multivariate Supervised OPLS-DA Models of Elicited T. baccata Suspension Cell Cultures and Determination of VIP Markers 

To efficiently determine the differences attributed to COR- and SA-mediated elicitation, a supervised orthogonal projection of latent structures discriminant analysis (OPLS-DA) was carried out for both elicitation treatments throughout the experiment (Figure 2). All OPLS-DA models were statistically validated by CV-ANOVA (*p* < 0.001), featuring their significance for discriminating the effect of elicitors on the phenolic profile of elicited cell suspensions at all experimental time points (days 8, 16, and 24) (Figure 2A,D,G, respectively). At day 8, the most differential profile was observed in COR-elicited samples according to the main orthogonal vector, while SA-elicited samples also differed from the control according to the secondary latent vector (Figure 2A). The effect of SA elicitation was most notable at day 16 as indicated by the discrimination associated to the main latent vector, whereas COR samples were separated from the control through the secondary vector (Figure 2D). Finally, at day 24, the net differences between the two types of elicited samples had decreased, while being clearly separated from the control by the major latent vector (Figure 2G), being in line with the results provided by the HCA. All models exhibited high-quality parameters in terms of goodness of fit (R^2^Y = 0.984, 0.990, and 0.990, for 8, 16, and 24 days, respectively) and goodness of prediction (Q^2^ = 0.969, 0.985, and 0.986, for 8, 16, and 24 days, respectively).

The supervised OPLS−DA models were combined with VIP analysis to detect which compounds contributed the most to the discrimination between treatments at each time point, known as VIP markers. For each VIP marker, the VIP score was combined with the logFC values of each treatment with respect to control. The full lists of VIP markers showing a VIP score > 1.0 and logFC > 2 or logFC < −2 with respect to control are provided in Appendix A for each time point (8, 16, and 24 days, respectively). Up to 34 different VIP markers were identified at day 8 (Appendix A), 41 at day 16 (Appendix A), and 36 at day 24 (Appendix A).

According to the results, at day 8 of elicitation (Figure 2B), the most abundant VIP markers (32.4%) were LMW polyphenols, followed by flavonoids (26.5%) and phenolic acids (23.5%), with lignans representing 14.7% and stilbenes only 2.9%. The most discriminant compounds according to their VIP score (Appendix A) were the phenolic acid 4-hydroxybenzoic acid 4-O-glucoside (VIP score = 1.37; logFC COR = −6.95; logFC SA = 2.87), followed by the phenolic terpene carvacrol (VIP score = 1.36; logFC COR = 0.00; logFC SA = 9.39), and the stilbene piceatannol (VIP score = 1.33; logFC COR = −0.21; logFC SA = −2.49). One lignan, cyclolariciresinol (VIP score = 1.24; logFC COR = −6.95; logFC SA = −9.39), and two flavonoids, the chalcone butein (VIP score = 1.11; logFC COR = −1.91; logFC SA = 0.06) and the flavanone naringenin 7-*O*-glucoside (VIP score = 1.10; logFC COR = −1.58; logFC SA = 0.01), were also among the 10 most discriminant phenols. Regarding the general modulation of VIP markers expressed as sum of logFC (Figure 2C), flavonoid biosynthesis was upregulated by both elicitors, especially by COR (sum of logFC > 15), being mainly represented by dihydroflavonoids, i.e., dihydromyricetin 3-*O*-rhamnoside (dihydroflavonol) and 3-hydroxyphloretin 2′-*O*-glucoside (dihydrochalcone). COR also stimulated the production of lignans, as shown for 1-acetoxypinoresinol, 7-hydroxymatairesinol, and anhydro-secoisolariciresinol, whereas SA slightly upregulated the biosynthesis of LMW polyphenols, mainly phenolic terpenes such as carvacrol and carnosic acid. In contrast, COR clearly downregulated the biosynthesis of LMW polyphenols and phenolic acids. Finally, piceatannol, the only stilbene VIP marker detected at day 8, was exclusively downregulated by SA elicitation (logFC = −2.49; Appendix A). 

At day 16 (Figure 2E), the proportions of phenolic groups identified as VIP markers changed, the most predominant being phenolic acids (31.7%) and flavonoids (26.8%). Additionally, lignans (19.5%) increased by almost 5% and LMW polyphenols (19.5%) clearly decreased by 13% with respect to day 8. Nevertheless, according to VIP scores (Appendix A), the most discriminant compounds were classified as other polyphenols, such as epirosmanol (VIP score = 1.14; logFC COR = 2.94; logFC SA = 2.90), 3,4-dihydroxyphenylglycol (VIP score = 1.14; logFC COR = 4.00; logFC SA = 0.00), and carnosol (VIP score = 1.14; logFC COR = 3.35; logFC SA = 1.35), or lignans, such as conidendrin (VIP score = 1.14; logFC COR = 4.00; logFC SA = 0.00) and 1-acetoxypinoresinol (VIP score = 1.14; logFC COR = 4.00; logFC SA = 0.00). Regarding the biosynthetic modulation of VIP markers (Figure 2F), both elicitors caused the downregulation of flavonoids. In detail, SA drove the down-accumulation of the flavanols prodelphinidin dimer B3, (+)-gallocatechin, and (+)-catechin (logFC = −13.2 for all of them). SA also exhibited a strong negative effect on the production of phenolic acids and lignans, while slightly stimulating the synthesis of hydroxylated forms of LMW polyphenols, including hydroxycoumarins (esculetin), hydroxybenzaldehydes (syringaldehyde and *p*-anisaldehyde), and hydroxyphenylpropenes (anethole). On the other hand, COR clearly upregulated the accumulation of lignans such as conidendrin, cyclolariciresinol, and lariciresinol-sesquilignan (logFC = 2.3−4.0 for all of them), as well as that of LMW phenolics carnosol and 3,4-dihydroxyphenylglycol (logFC = 3.35 and 4.00, respectively). Interestingly, the biosynthesis of the stilbene resveratrol 3-O-glucoside (VIP score = 1.03) was strongly induced by SA (logFC = 13.19) but not affected by COR (logFC = 0.00), with respect to control (Appendix A).

At the end of the experiment, day 24 (Figure 2H), the most representative VIP markers were phenolic acids (33.3%) and flavonoids (25%). The proportion of stilbenes (5.6%) had doubled compared to day 16 and LMW polyphenols (27.8%) had also increased, whereas lignans (8.3%) suffered a decrease of 11.2%. Again, other polyphenols and lignans showed the highest VIP scores (Appendix A), as observed for the hydroxycoumarin esculetin (VIP score = 1.20; logFC COR = 2.05; logFC SA = 1.09), followed by 3,4-dihydroxyphenylglycol (VIP score = 1.20; logFC COR = 4.21; logFC SA = 0.25) and the lignan conidendrin (VIP score = 1.18; logFC COR = 2.98; logFC SA = 1.79). Regarding the net accumulation trends of VIP markers (Figure 2I), both elicitors drove a downregulation effect on all the phenolic families, especially phenolic acids and flavonoids, with respect to the untreated control. The only compounds showing an increased production at day 24 were LMW polyphenols in COR-elicited samples.

The time-dependent distribution of VIP markers was reported though a Venn diagram (Figure 3), which indicates that 13 phenolic compounds were found at all experimental time points in both elicitation treatments, being represented by hydroxycinnamic acids (caffeic acid 4-*O*-glucoside, ferulic acid 4-*O*-glucoside, and 3-*p*-coumaroylquinic acid); flavonoids, including chalcones (butein), flavones (gardenin B and tetramethylscutellarein), and flavanones (6-prenylnaringenin); lignans (episesaminol and 7-hydroxymatairesinol); and other polyphenols (hydroxycoumarins and the furanocoumarin bergapten). A high number of compounds (12) were detected both at days 16 and 24, whereas only 6 compounds were commonly spotted at days 8 and 16. These results indicate that both the duration and type of elicitation significantly influenced the metabolic fingerprint of *T. baccata* cell suspensions. According to the OPLS-DA models, COR induced an earlier reprogramming of phenolic metabolism than SA and, overall, provoked a greater downregulation of LMW polyphenol and phenolic acid biosynthesis. At day 8, COR elicitation was observed to stimulate the production of flavonoids and lignans, which was reverted from day 16, when it drove the upregulation of lignan biosynthesis while suppressing that of phenolic acids. The elicitor SA initially stimulated the accumulation of flavonoids and LMW polyphenols, although the former subsequently decreased over time. Its effect on the production of lignans and phenolic acids throughout the experiment was either negligible or strongly suppressive. Finally, SA was the only elicitor able to actively upregulate the biosynthesis of VIP marker stilbenes, although only at day 16.

### 3.3. Semi-Quantification of the Phenolic Compounds in Elicited T. baccata Suspension Cell Cultures 

The results for the semi-quantification of phenolic compounds in the elicited *T. baccata* cell suspensions are shown in Figure 4. As a general rule, the highest content of all phenolic families was found in the elicited samples, mainly those treated with COR at days 8 and 16, with the exception of stilbenes. The highest flavonoid content was ~200 µg LE/gDW, achieved by COR elicitation, followed by almost ~150 µg LE/g DW in SA-elicited samples, both at day 8 (Figure 4A). However, at day 16, levels had decreased significantly, being 2.2-fold lower in COR and 11-fold lower in SA treatments compared to the control. Similarly, at the end of the experiment, the flavonoid content in control samples was 4.6-fold and 3.5-fold higher with respect to COR and SA conditions, respectively. The highest amount of lignans, ~300 µg SE/g DW, was also detected at day 8 in COR-treated samples, being 3-fold higher with respect to control and 1.7-fold higher compared with SA elicitation (Figure 4B). At day 16, no lignans were detected in SA-elicited samples, whereas the amounts in control and COR samples were similar. Finally, at day 24, the highest amount (~80 µg SE/g DW) was found in the control. The highest production of LMW polyphenols throughout the experiment was reported in COR-elicited cell cultures throughout all the time points: ~1000 µg TE/g DW (Figure 4C). In contrast, in SA-elicited samples, levels decreased steadily over time, although they remained higher than in control conditions until day 24, when they were around 1.58-fold lower. The phenolic acid content was highest at day 8 under COR elicitation (~1200 µg FE/g DW), although similar rates were achieved in the control at day 24 (~1000 µg FE/g DW). In SA-elicited cultures, the maximum production was at days 8 and 24 (~800 µg ferulic acid/g DW). Surprisingly, in the case of stilbenes, the maximum content of ~65 µg RE/g DW was observed after 24 days of culture in untreated cell suspensions (Figure 4E). At this time point, the amount was 3.65-fold and 4.33-fold times higher than in COR and SA conditions, respectively. However, at day 8, the highest content was detected in COR-elicited cultures (~25 µg *trans*-resveratrol/g DW) and at day 16 in SA-elicited cell suspensions (~18 µg *trans*-resveratrol/g DW).

Considering the combination of all phenolic families, the maximum total phenolic content (TPC) (Figure 5) was achieved at day 8 in COR−elicited samples (almost 3000 µg/g DW), which was 2−fold and 1.5−fold higher compared to control and SA-elicited samples, respectively. However, at day 16, the TPC of COR samples was similar to that of control cultures, having undergone a 1.5−fold reduction. In the same way, SA-elicited samples also showed a decrease in TPC over time (~1343 µg/g DW at day 16). Surprisingly, the second highest TPC of the whole experiment (~2243 µg/g DW) was observed in control conditions at day 24, being 1.25−fold and 1.47-fold higher than in COR- and SA−elicited samples, respectively.

In general, these results show that elicitation treatments represent an effective technique for increasing the production of all the studied phenolic families, except for stilbenes, whose synthesis was more dependent on time than the action of elicitors. COR proved to be stronger and faster acting than SA, as the highest amounts of all the polyphenols were found near the beginning of the experiment, at day 8 in COR-elicited cultures. Its impact varied according to the type of phenolic compound, being short-lived in the case of flavonoids and lignans, which underwent a marked steady decrease after the initial increase, and more constant over time for LMW polyphenols and phenolic acids. 

### 3.4. Effect of Elicitation on Cell Biomass of Taxus baccata Cell Suspension Cultures

Besides secondary metabolism, elicitation also affected other cell parameters, such as cell growth, as measured by both fresh weight (FW) and dry weight (DW) (Figure 6). In the FW time course (Figure 6 Top), after a stagnation at the beginning of the experiment, cell suspensions cultured in control conditions experienced an increase of approximately 17% in relation to the inoculum, reaching a maximum of 350 g/L. In contrast, the elicitation of cultures resulted in a significant decrease in the FW, with the effects of COR and SA not differing significantly until day 24, when a decrease of 20% and 13%, respectively, was observed. The differences between the elicitors were more obvious according to DW values (Figure 6 Bottom), COR inhibiting cell growth more strongly, and SA not differing significantly from the control conditions until day 24. At that time point, SA-elicited cells showed a decrease in DW of 7.14%, and COR-treated cells, 14.29%. A slight reduction in cell growth was observed in untreated cell suspensions, which may be attributed to the stress suffered by cells during adaptation to the new fresh medium. The negative impact of the elicitors on cell growth could be due to the transition from primary metabolism, involved in biomass formation and developmental functions, to specialized metabolism, which is focused on stress tolerance and the biosynthesis of secondary metabolites such as polyphenols. 

### 3.5. Determination of Antioxidant Activity by DPPH Radical Scavenging Assay

The DPPH radical scavenging assay is currently one of the most widely used methods for determining the antioxidant capacity of a sample due to its speed, simplicity, and accuracy. DPPH assay assesses free radical scavenging activity by reduction of DPPH to DPPH-H through donating hydrogen atoms or electrons. In the present study, the DPPH method was selected for determining the antioxidant capacity of *T. baccata* cell extracts in control and elicited conditions either with 1 µM of COR or with 150 µM of SA at all the experimental time points: days 8, 16, and 24 (Figure 7). The EC_50_ values of the samples were determined and compared to the positive control, Trolox, which had an EC_50_ = 47 µg/mL. The % inhibition ratio of each sample was plotted at different concentrations of methanolic cell extract, ranging from 10 µg/mL to 100 µg/mL. Then, the regression lines were used to calculate the 50% inhibition ratio of each sample. The SA treatment showed a low profile of EC_50_ values along time, reaching the minimum value (141 µg/mL ± 12.8) at 24 days. The samples of COR treatment showed significant increments along the days and the lowest EC_50_ = 138 µg/mL ± 9.2 for all treatments and days. The control had the highest increments along the days. The SA treatment exhibited a low and relatively stable profile of EC_50_ values over time, with the minimum value of 141 µg/mL ± 12.8 observed at 24 days. On the other hand, the COR treatment showed a significant increase in EC_50_ values over time, with the lowest value at day 8 of 138 µg/mL ± 9.2, which also represents the lowest value observed for all treatments and days. The control treatment exhibited the highest increase in EC_50_ values over time, where the minimum value was 153 µg/mL ± 11.1 and the maximum 410 µg/mL ± 26.7. Overall, the samples demonstrated the highest antioxidant activity at day 8. Despite our better result being almost three times less effective than Trolox, according to the EC_50_ value, it is important to consider the sustainability of the source. While Trolox is a purified compound, obtaining it can require significant resources and have negative environmental impacts. In contrast, the natural antioxidants from *T. baccata* cells offer a more sustainable alternative as they come as waste from paclitaxel production systems.

## 4. Discussion

The impact of elicitation with either 1 µM of coronatine (COR) or 150 µM of salicylic acid (SA) on the metabolic fingerprint of *T. baccata* cell suspension cultures was studied using an untargeted metabolomics approach. This combinatorial methodology was applied to explore the sustainability of *Taxus* cell cultures, well known as producers of the anticancer drug paclitaxel, which could also serve as a source of valuable antioxidant phenolic compounds for an efficient optimization of resources.

Paclitaxel, a diterpene alkaloid, is widely used for the treatment of various types of cancer due to its chemotherapeutic properties. The cost-effectiveness and controlled growing conditions of optimized *Taxus* spp. cell cultures make this biotechnological system the current method of choice for paclitaxel production [23]. The aim of the present research was to increase the environmental sustainability of in vitro *Taxus* cultures by reusing the raw material to obtain phenolic compounds to be potentially applied as natural antioxidants. As a convenient analytical approach to reveal the effect of elicitation on the phenolic profile of plant cell cultures, we used untargeted metabolomics analysis. In the case of elicited *T. baccata* cell suspension cultures, the application of this approach provided a total of 83 phenolic compounds, which were classified into 5 families: phenolic acids; lignans; stilbenes; LMW polyphenols, including alkylphenols, coumarins, phenylpropenes, quinones, and tyrosols; and flavonoids, including flavones, flavonols, anthocyanins, flavanols, flavanones, isoflavonoids, and dihydrochalcones. To the best of our knowledge, there has been no previous report on an exhaustive profiling of phenolic metabolites in a *Taxus* species, which includes an analysis of post-elicitation variations in phenolic biosynthesis over time.

In general terms, both elicitors modulated the biosynthesis of polyphenols, reaching the highest phenolic content by a short-term COR−based elicitation. Accordingly, multivariate analyses indicate a time−dependent response of the tested elicitors, since the most notable changes in phenolic metabolism due to COR elicitation were reported after 8 days, whereas SA−driven elicitation achieved the maximum modulation after 16 days. Such a modulatory response was found differential according to the different families of phenolic compounds. In the case of flavonoids and lignans, COR elicitation induced an initial increase followed by a marked, steady decrease over time, whereas it induced a sustained enhancing effect on LMW polyphenols and phenolic acids. Another culture parameter affected by the elicitation treatments was cell growth. Whereas *T. baccata* cells exhibited a relatively good growth in the optimized medium for taxane production [24], biomass formation was limited by SA and even more so by COR, as observed in previous *Taxus* experiments [17,25,26]. The earlier and more efficient metabolic response to COR compared to that of SA may be attributed to its chemical structure, which mimics the isoleucine-conjugated form of jasmonic acid (JA-Ile), while having a more stable and rigid cis−orientation in its bicyclic skeleton. As a result, it cannot be transformed into less active forms by cytochrome P450 (CYP94B3), the enzyme involved in JA-Ile turnover, and consequently does not lead to attenuation of JA-induced signalling [27,28]. 

Regarding metabolic variations over time, up to 34 VIP compounds were detected at day 8, 41 at day 16, and 36 at day 24 of culture. According to the literature, the biosynthetic modulation of these metabolites by elicitation may be due to their antioxidant activity, as phenylpropanoids have been shown to develop an effective response to the oxidative burst triggered by elicitor-induced stress [29]. The high redox reactivity of phenolic compounds reduces free radicals and prevents destructive cascade reactions, thus becoming a highly effective defensive mechanism at the intracellular level [29]. Moreover, phenolic compounds are of interest for their beneficial effects on human health, as they are associated with a capacity to reduce the development of chronic cardiovascular and neurodegenerative diseases. Their bioactivities include strengthening of blood vessel walls, improvement of digestion, reduction of blood lipids, stimulation of human immunity, and prevention of bacteria and cancer cell proliferation [30]. Previous studies have explored the *Taxus* genus as a source of these natural antioxidants. For example, [7] detected six phenolic acids, including caffeic acid, chlorogenic acid, and protocatechuic acid, in bark extracts of *T. cuspidata*, which showed antiproliferative activities against Jurkat, MCF-7, and HeLa cancer cells. The authors of [8] performed a phytochemical characterization of the total phenolic, flavonoid, and carotenoid contents of *T. baccata* arils and confirmed the antiproliferative and proapoptotic activity of rhodoxanthin, the main carotenoid found in this fruit. In contrast, few such studies have been carried out with *Taxus* cell suspensions, even though plant cell cultures are being used as production systems of a wide range of valuable secondary metabolites, such as polyphenols, alkaloids, terpenes, and lignans [30]. For example, [31] reported that the in cellulo antioxidant activity of purple basil calli (*Ocimum basilicum* L. var *purpurascens*) was enhanced by the elicitor-induced production of phenolic compounds. In addition, other authors also postulated the link between the phenolic content and antioxidant potential [32,33]. Our results support the previous statement, as we observed that the elicited samples with the highest total phenol content also exhibited the highest percentages of inhibition in the DPPH assay. This finding indicates that the phenolic compounds present in *T. baccata* cell extracts contribute significantly to their antioxidant activity.

The most abundant phenolic compounds in the *T. baccata* cell cultures throughout the experiment were phenolic acids and LMW polyphenols, followed by lignans, flavonoids, and stilbenes. The maximum total phenolic content was achieved at day 8 in COR−elicited samples (almost 3000 µg/g DW) compared to day 24 in control conditions (2243.5 µg/g DW). Among the phenolic acids, the highest content of hydroxycinnamic acids was obtained at day 8 after COR elicitation; notably, the content of ferulic acid (approximately 1200 µg/g DW) was 2-fold higher than in control conditions. Similarly, [34] reported the beneficial effect of methyl jasmonate (MeJA) on the accumulation of phenolic acids, mainly rosmarinic acid, in in vitro *Salvia bulleyana* hairy roots; the total content doubled compared to the control after only 3 days of exposure to the elicitor. The antioxidant, anticancer, and antimicrobial properties of this class of phenolic compounds were also clearly demonstrated, their scavenging activity against different free radicals being 3−fold higher compared to industrial synthetic antioxidants. Additionally, in a study about the role of elicitation in alleviating drought stress in *Thymus lotocephalus* in vitro cultures, [35] found that MeJA was more effective than SA in promoting the biosynthesis of phenolic compounds (mainly phenolic acids), increasing their accumulation by up to 38.8%. 

The highest production of LMW phenolics was observed in COR−elicited cultures throughout the experiment, the levels always being above or equal to 1000 µg TE/g DW, with a peak at day 8. In contrast, in SA−elicited cultures, the LMW polyphenol content decreased steadily over time. A positive effect of elicitation on LMW phenolics has been reported previously. The authors of [30] tested a strategy that combined elicitation with 100 µM of SA or 100 µM of MeJA with precursor feeding in cell suspensions of *Cannabis sativa* during 9 days of culture. The best results were obtained when using MeJA together with tyrosine feeding, which led to the accumulation of 2.19 µg tyrosol/mg DW after only 4 days of culture, representing a 3.32-fold increase compared to the control. 

Lignans were also putatively annotated in the *T. baccata* cell cultures, and the highest content was observed at day 8 of COR treatment for sesamin equivalents (about 300 µg SE/g DW). Different types of lignans have been reported previously in *Taxus* spp. For example, [5] identified and characterized the phytochemical structure of (+)-taxiresinol, (+)-isotaxiresinol, (+)-isolariciresinol, and (-)-secoisolariciresinol from the wood and roots of *T. cuspidata* and also confirmed their in vitro antioxidant and radical-scavenging activities. Further, [6] isolated five lignans from the heartwood of *T. baccata* and demonstrated their positive role in inhibiting butyrylcholinesterase (BChE) and lipoxygenase (LOX) activities, which play a role in the pathogenesis of Alzheimer’s disease. Lignans have also been postulated to exhibit antitumor, antiviral, hepatoprotective, anti-inflammatory, and antimicrobial activities, among other beneficial properties [36]. However, to date, a few studies such as [15] have explored the biosynthetic potential of a *Taxus*-based in vitro system, namely two MeJA-elicited *Taxus* × *media* hairy root lines, for the production of lignans of therapeutic interest. The results of the present research confirm the suitability of *Taxus* plant cell cultures to be exploited as a biofactory of this family of polyphenols. 

Finally, although flavonoids and stilbenes were the least representative families in elicited *T. baccata* cell cultures, they were also identified as VIP markers throughout the experiment. Like other families, the highest flavonoid content was achieved at day 8 in elicited cultures (200 µg LE/g DW for COR and 150 µg LE/g DW for SA). Notably, stilbenes were the only phenolic group whose levels were not improved by elicitation; for example, the maximum content of 65 µg RE/g DW was observed at day 24 of culture in untreated cell suspensions. This may indicate that stilbene biosynthesis in *Taxus* spp. is more dependent on time than the action of elicitors.

## 5. Conclusions

The comprehensive profiling of phenolic metabolites in a *Taxus* species, including an analysis of post-elicitation variations in phenolic biosynthesis over time, presented in this work allowed us to visualize the specific effects and distribution of polyphenols under different proposed treatments. The best antioxidant activity results were observed in the cell cultures treated with COR, which is also a potent elicitor for the production of taxanes, which are collected prior to the cell extraction proposed in this research. The treatment with SA also demonstrated good antioxidant activity results, surprisingly showing a more consistent profile over time. Additionally, cluster and multivariate analyses revealed how different groups of phenolic compounds can achieve similar antioxidant responses. Our results suggest that cell culture waste from paclitaxel production systems can be repurposed as a natural source of antioxidants for the pharmaceutical and/or food industry. This approach to valorizing *Taxus* culture residues has the potential to lead to the development of novel applications.

## Figures and Tables

**Figure 1 antioxidants-12-00887-f001:**
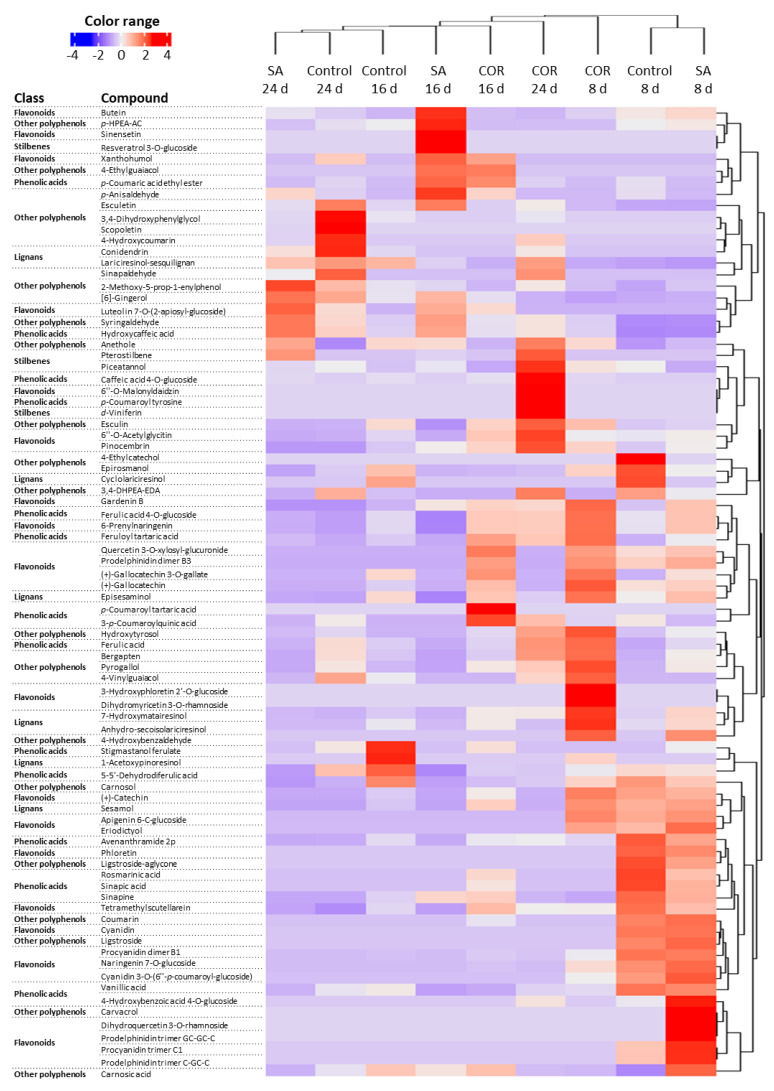
Unsupervised hierarchical cluster analysis on extracts from *T. baccata* cell suspensions either untreated (control) or elicited with 1 µM coronatine (COR) or 150 µM salicylic acid (SA), at different experimental time points (days 8, 16, and 24). The heatmap is based on the average values of phenolic compounds, and the data were preprocessed by autoscaling (unit variance scaling, UV). Clustering was performed according to averages algorithm and based on Euclidean distances.

**Figure 2 antioxidants-12-00887-f002:**
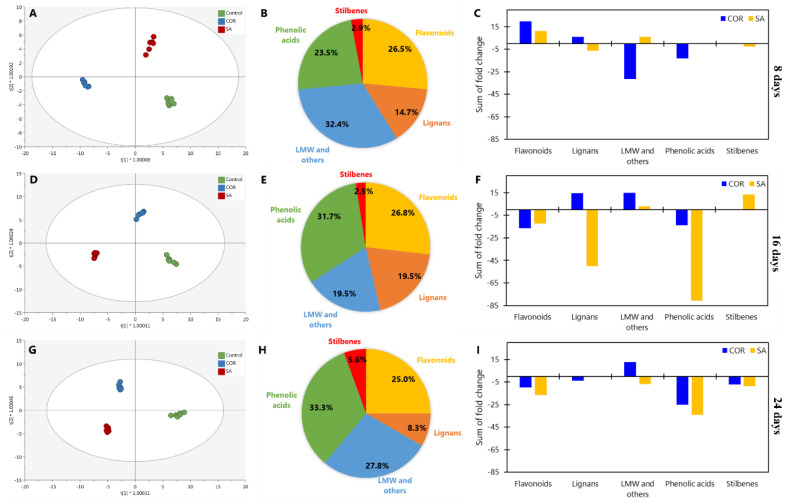
Chemometrics analysis based on multivariate supervised orthogonal projection to latent structures discriminant analysis (OPLS−DA) combined with variable importance in projection (VIP) analysis on extracts from *T. baccata* cell suspensions either untreated (control) or elicited with 1 µM coronatine (COR) or 150 µM salicylic acid (SA) at the following experimental time points: (**A**–**C**) 8 days; (**D**–**F**) 16 days; and (**G**–**I**) 24 days. The OPLS-DA models showed high goodness of fit and predictability (**A**,**D**,**G**). The VIP markers related to each model are provided in Appendix A and are depicted according to their phenolic class and proportion (**B**,**E**,**H** respectively). The modulation of each phenolic class by the elicitor treatments is expressed as the sum of logFC of individual compounds belonging to the same class (**C**,**F**,**I**).

**Figure 3 antioxidants-12-00887-f003:**
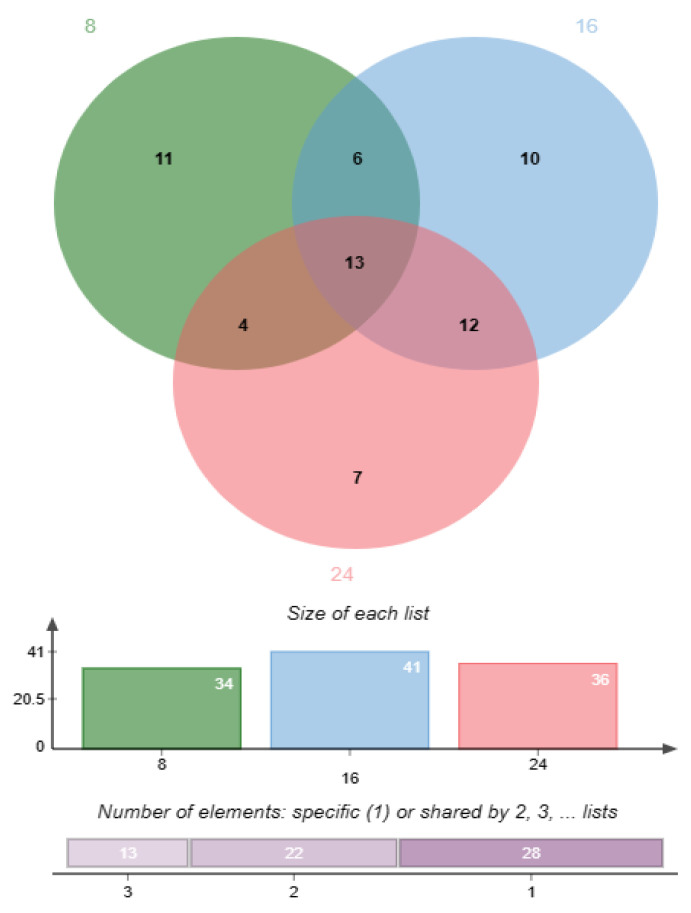
Venn diagrams showing the variable importance in projection analysis (VIP) markers (FC ≥ 2; VIP score > 1) in extracts from *T. baccata* cell suspensions either untreated (control) or elicited with 1 µM coronatine (COR) or 150 µM salicylic acid (SA): at each experimental time point (days 8, 16, and 24) in both elicited cultures. The complete lists of VIP markers are provided in Appendix A.

**Figure 4 antioxidants-12-00887-f004:**
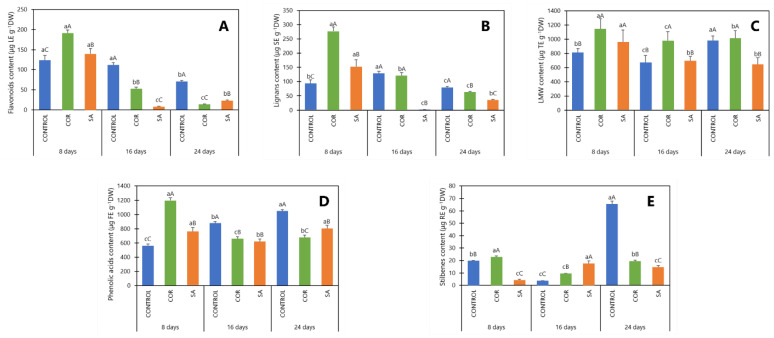
Semi−quantification of phenolic families in the extracts from *T. baccata* cell suspensions either untreated (control) or elicited with 1 µM coronatine (COR) or 150 µM salicylic acid (SA) at all the experimental time points (days 8, 16, and 24). (**A**) Flavonoid content expressed as µg of luteolin equivalents per gram of dry weight (µg LE/gDW). (**B**) Lignan content expressed as µg of sesamin equivalents per gram of dry weight (µg SE/gDW). (**C**) Low−molecular−weight polyphenol content expressed as µg of tyrosol equivalents per gram of dry weight (µg TE/gDW). (**D**) Phenolic acid content expressed as µg of ferulic acid equivalents per gram of dry weight (µg FE/gDW). (**E**) Stilbene content expressed as µg of *trans*−resveratrol equivalents per gram of dry weight (µg RE/gDW). Vertical bars indicate standard deviation (n = 6). Lowercase letters indicate significant differences (*p* < 0.05) within the same treatment between experimental time points. Capital letters indicate significant differences (*p* < 0.05) between treatments within the same experimental time point. The statistical differences were evaluated according to Tukey’s test.

**Figure 5 antioxidants-12-00887-f005:**
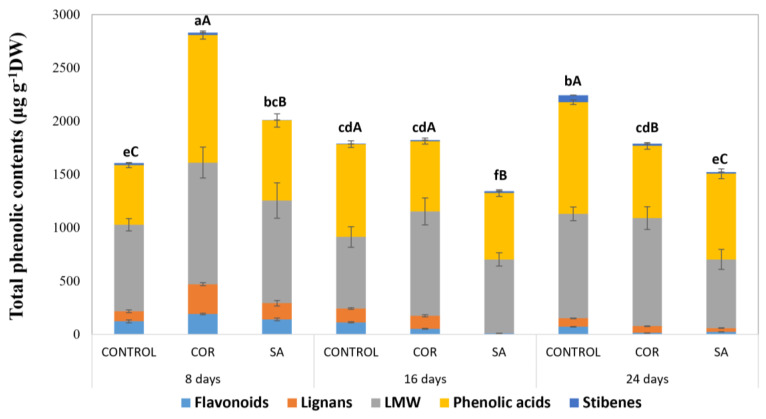
Semi−quantification of the total phenolic content (TPC) in the extracts from *T. baccata* cell suspensions either untreated (control) or elicited with 1 µM coronatine (COR) or 150 µM salicylic acid (SA) at all the experimental time points (days 8, 16, and 24). Flavonoid content expressed as µg of luteolin equivalents per gram of dry weight (µg LE/g DW). Lignan content expressed as µg of sesamin equivalents per gram of dry weight (µg SE/g DW). Low−molecular−weight phenolic content expressed as µg of tyrosol equivalents per gram of dry weight (µg TE/g DW). Phenolic acid content expressed as µg of ferulic acid equivalents per gram of dry weight (µg FE/g DW). Stilbene content expressed as µg of *trans*−resveratrol equivalents per gram of dry weight (µg RE/g DW). Vertical bars indicate standard deviation (n = 6). Lowercase letters indicate significant differences (*p* < 0.05) within treatments between experimental time points. Capital letters indicate significant differences (*p* < 0.05) between treatments within the same experimental time point. The statistical differences were evaluated according to Tukey’s honestly significant different test.

**Figure 6 antioxidants-12-00887-f006:**
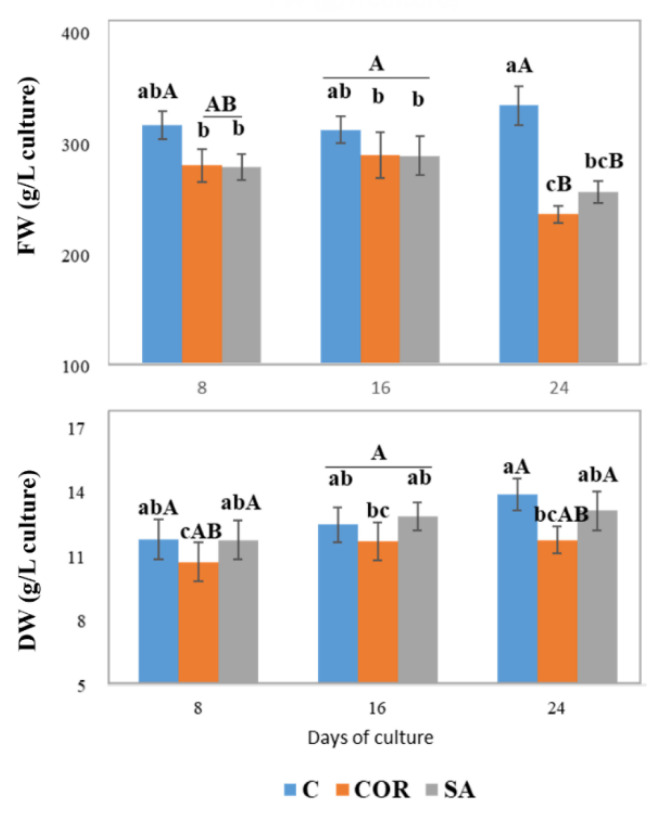
Time courses of biomass production (measured as fresh and dry weight) of *T. baccata* cell suspensions cultured for 24 days in the production medium in control conditions (C) or with the addition of 1 µM COR (coronatine) or 150 µM SA (salicylic acid). In all cases, the inoculum consisted of 300 g/L of cells. Data represent average values from three separate experiments ± SD. Lowercase letters indicate significant differences (*p* < 0.05) within treatments between experimental time points. Capital letters indicate significant differences (*p* < 0.05) between treatments within the same experimental time point. The statistical differences were evaluated according to Tukey’s honestly significant different test.

**Figure 7 antioxidants-12-00887-f007:**
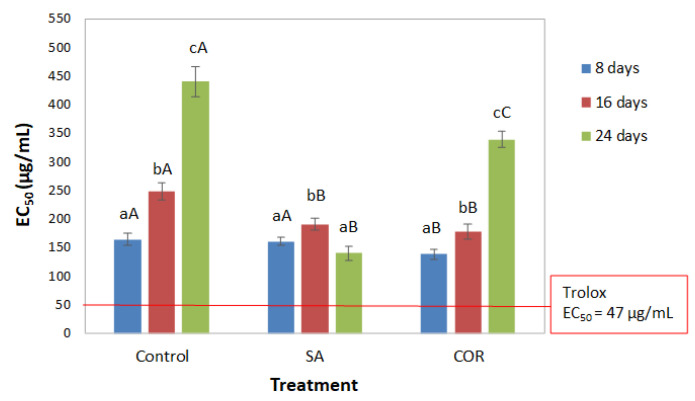
Determination of antioxidant activity of *T. baccata* cell suspensions by DPPH radical scavenging assay expressed as EC_50_ value. The cell suspensions were cultured either in control conditions or elicited with 1 µM coronatine (COR) or 150 µM salicylic acid (SA) at all the experimental time points (days 8, 16, and 24). Data represent average values from three separate experiments ± SD. Lowercase letters indicate significant differences (*p* < 0.05) within treatments between experimental time points. Capital letters indicate significant differences (*p* < 0.05) between treatments within the same experimental time point. The statistical differences were evaluated according to Tukey’s honestly significant different test.

## Data Availability

Data are contained within the article and Appendix A.

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
