# Peer review of "Impact of Elicitation on Plant Antioxidants Production in *Taxus* Cell Cultures"

_antioxidants, 2023, doi:10.3390/antiox12040887_

Round 1

Reviewer 1 Report

In the present manuscript, the effect of elicitation on the synthesis of bioactive compounds in cell cultures of Taxus bacata was studied. For this purpose, two elicitors were tested, e.g. 1 μM coronatine and 150 μM salicylic acid.

Line 78: correct to "biofactories".

Lines 84, 97: add the name of authors of the reference. Check throughout the text for similar mistakes.

Figure 1 does not provide any legible information for the reader. It is not clear which compounds increase or decrease over time.

Figure 4. Use the same color for the bars of the same treatments. 

Figure 5. A two way ANOVA should be performed as in Figure 4. The same applies to Figures 6 and 7. 

The yield of compounds per culture sghould be discussed in section 3.5

Author Response

RESPONSE: Thank you for reading our paper and helping with your comments to improve it. Below is a point-by-point entry of the changes made. Additionally, in the manuscript, we highlight the changes made in yellow.

Line 78: correct to "biofactories".

Typo corrected.

Lines 84, 97: add the name of authors of the reference. Check throughout the text for similar mistakes.

References corrected.

Figure 1 does not provide any legible information for the reader. It is not clear which compounds increase or decrease over time.

We have remade figure 1, this time we include legible information about the compounds and their behavior along the treatments. Additionally, we have enhanced the clarity and coherence of the description of these results to ensure that they are easily comprehensible to the reader.

Figure 4. Use the same color for the bars of the same treatments. 

We have changed the color.

Figure 5. A two way ANOVA should be performed as in Figure 4. The same applies to Figures 6 and 7. 

The two-way ANOVA was performed for the indicated figures and the legend of each one has been edited.

The yield of compounds per culture sghould be discussed in section 3.5

In the new version of the manuscript, this section 3.5 was adapted as reviewer 3's suggestion to express antioxidant activity as EC50 values. This new way of showing the results allows us to know how much quantity by weight (of cells) is necessary to generate an antioxidant effect. The corresponding discussion of these results is included in this section 3.5 and we hope this answers your comment.

Reviewer 2 Report

The review of “Searching for new biosustainable sources of plant antioxidants: the effect of elicitation on polyphenol production in Taxus cell cultures” for Antioxidants MDPI. The topic of manuscript fits within the scope of the journal and results can be considered of interest in order to produce valuable phenolic antioxidants by using Taxus plant cell.

The experiments appear to have been correctly designed and performed, the methods used are appropriate, and the manuscript presents a considerable amount of experimental results, which are in general coherent, and clearly presented, described and discussed. Furthermore, the discussion of the work are sound and conclusions are based on the results presented in the manuscript. Therefore, in my opinion, no serious criticisms can be raised on this study.

I have just a few remarks, which I give under author’s consideration:

1. The title of manuscript should be corrected. In present version, the title does not reflect the experimental study and sounds like title of review paper.

2. Figure 2 (C, F, and I): Please check the presentation of this figure. I assume that legends “8 days”, “16 days” and 24 days” overlap part of the graphic information.

3. Please, add to the legend to Figure 4 what statistical test was used to check significance of difference.

4. The terms “LMW phenolics” and “LMW polyphenols” are  confused. Usually, phenolic acids and flavonoids are classified as LMW phenolics. Please add an explanation based on what criteria you have allocated compounds to this group.

5. Discussion, L 527-530: Please report results of correlation analysis when mentioning it in text. What is the P-Value for correlation? What test was performed to determine it?

Author Response

RESPONSE: Thank you for reading our paper and helping with your comments to improve it. Below is a point-by-point entry of the changes made. Additionally, in the manuscript, we highlight the changes made in yellow.

  1. The title of manuscript should be corrected. In present version, the title does not reflect the experimental study and sounds like title of review paper.

We modified the title of our manuscript to reflect better the experimental study. The new proposed title is:

Impact of elicitation on plant antioxidants production in taxus cell cultures.

  1. Figure 2 (C, F, and I): Please check the presentation of this figure. I assume that legends “8 days”, “16 days” and 24 days” overlap part of the graphic information.

We edited figure 2 to avoid overlapping graphic information.

  1. Please, add to the legend to Figure 4 what statistical test was used to check significance of difference.

We have updated the legend of Figure 4 to include information about the statistical test.

  1. The terms “LMW phenolics” and “LMW polyphenols” are  confused. Usually, phenolic acids and flavonoids are classified as LMW phenolics. Please add an explanation based on what criteria you have allocated compounds to this group.

We included an explanation in section 2.4 to clarify the use of LMW term in the present work.

  1. Discussion, L 527-530: Please report results of correlation analysis when mentioning it in text. What is the P-Value for correlation? What test was performed to determine it?

We couldn’t find in the L527-530 section any reference to correlation analysis. The paragraph in this section is more a statement of causation or a proposed mechanism. We revised the manuscript and clarified the text to avoid confusion about any correlation analysis. Additionally, we include a new section in material and methods to specify the statistical parameters and tests used in this work.

Reviewer 3 Report

The paper aimed to study the effect of two elicitors, i.e. coronatine (1 μM) and salicylic acid (150 μM) on the polyphenolic profile (UHPLC/QTOF-MS method) and antioxidant activity (DPPH assay) of Taxus baccata cell suspensions, with the use of combinatorial approach based on elicitation and untargeted metabolomics.

The idea of the work seems to be interesting, especially since it is the first report on such a comprehensive analysis of the qualitative composition of a Taxus baccata species. The manuscript is well-written. The findings are also well discussed. The experimental part lacks, however, some data important for the repetition of the work and verification of its scientific soundness (lack of positive control in the DPPH antioxidant activity test). In my opinion, the submission requires extensive edition and improvement at many points, the main of which are listed below.

Major issues:

L438-461: In the DPPH method, the Authors did not test antioxidant activity for the positive control, e.g. for natural compounds such as ascorbic acid or quercetin, or synthetic compounds such as Trolox or BHT, which is unacceptable. In addition, the antioxidant activity in the DPPH test should be expressed as EC50 values, both for the tested T. baccata cell extracts, control, and the positive control. Figure 7 should therefore present the EC50 values for the cell extracts (days 8, 16, 24) and the positive control.

Minor issues:

L127-128: please add the description of the gradient in the UHPLC method according to the scheme: e.g. “The elution system consisted of solvent A (0.5% water solution of orthophosphoric acid, w/v) and solvent B with the elution profile as follows: 0-1 min, 5% B (v/v); 1-4 min, 5-15% B; 4-10 min, 15% B; 10-11 min, 15-50% B; 11-15 min, 50% B; 15-16 min, 50-5% B; 16-22 min, 5% B (equilibration)”.

L135, 289, 434, 459, 558: please eliminate unnecessary spaces in the text of the manuscript. Please check the whole manuscript thoroughly.

L153: should be H2O.

L152-165: in the description of the methodology for the analysis of antioxidant activity, please add which compound was tested as a positive control.

L186-192: please consider this section as a separate subsection (Statistical analysis).

L300: the letter “p” in p-anisaldehyde should be written in italics.

L381 and 402: Authors should decide whether the word "trans" in the names of phenolic compounds should be italicized and should use this scheme throughout the text of the entire manuscript and Tables S1-S3.

L579-582: This fragment of the text should be presented as a separate section (Conclusions) and more substantively developed.

Please add in the Supplementary section:

11. one figure (Figure S1) presenting a UHPLC chromatogram of T. baccata cell suspension at e.g. 280 nm wavelength, on which all peaks of phenolic compounds (1-83) identified and described in Tables S1 will be marked.

22. please indicate in Table S1 which polyphenolic compounds were identified for the first time (de novo), as claimed by the Authors (L483).

Author Response

RESPONSE: Thank you for reading our paper and helping with your comments to improve it. Below is a point-by-point entry of the changes made. Additionally, in the manuscript, we highlight the changes made in yellow.

Major issues:

L438-461: In the DPPH method, the Authors did not test antioxidant activity for the positive control, e.g. for natural compounds such as ascorbic acid or quercetin, or synthetic compounds such as Trolox or BHT, which is unacceptable. In addition, the antioxidant activity in the DPPH test should be expressed as EC50 values, both for the tested T. baccata cell extracts, control, and the positive control. Figure 7 should therefore present the EC50 values for the cell extracts (days 8, 16, 24) and the positive control.

We made the modification to fix this major issues, editing the text of the 2.5 section and making a new figure 7. Figure 7 now shows the EC50 values for all the days and treatments. Also, we included the EC50 value for Trolox to show the reader a relevant comparison.

 Minor issues:

L127-128: please add the description of the gradient in the UHPLC method according to the scheme: e.g. “The elution system consisted of solvent A (0.5% water solution of orthophosphoric acid, w/v) and solvent B with the elution profile as follows: 0-1 min, 5% B (v/v); 1-4 min, 5-15% B; 4-10 min, 15% B; 10-11 min, 15-50% B; 11-15 min, 50% B; 15-16 min, 50-5% B; 16-22 min, 5% B (equilibration)”.

We clarified this section to fit better your scheme.  The new version show the next text: Elution was achieved by applying a continuous linear gradient, employing a binary mobile phase system that consisted in a mixture of acetonitrile (solvent A) and water (solvent B) for 32 minutes, following the profile: 0-32 min, 94%B – 6%B. An equilibration period of 3 min was employed to return to the initial elution condition prior the subsequent injection.

L135, 289, 434, 459, 558: please eliminate unnecessary spaces in the text of the manuscript. Please check the whole manuscript thoroughly.

We eliminated the unnecessary spaces in the text.

L153: should be H2O.

Typo corrected

L152-165: in the description of the methodology for the analysis of antioxidant activity, please add which compound was tested as a positive control.

We added this information in the Material and Methods section.

L186-192: please consider this section as a separate subsection (Statistical analysis).

We have included this new section.

L300: the letter “p” in p-anisaldehyde should be written in italics.

Typo corrected

L381 and 402: Authors should decide whether the word "trans" in the names of phenolic compounds should be italicized and should use this scheme throughout the text of the entire manuscript and Tables S1-S3.

Typo corrected

L579-582: This fragment of the text should be presented as a separate section (Conclusions) and more substantively developed.

We included the section and improve the content to sustain better the conclusions of the research.

Please add in the Supplementary section:

  1. one figure (Figure S1) presenting a UHPLC chromatogram of T. baccatacell suspension at e.g. 280 nm wavelength, on which all peaks of phenolic compounds (1-83) identified and described in Tables S1 will be marked.

The metabolomics-based approach employed in the present work is based on ultra-high performance liquid chromatography coupled to quadrupole-time-of-flight mass spectrometry (UHPLC/QTOF-MS). As indicated in section 2.4. The determination of compounds was made according to their mass spectrum and m/z profile assisted by bioinformatic tools without referring the UV-visible absorbance of compounds. In this sense, the chromatogram incorporated as Supplementary Figure S1 represents the total ion chromatogram (TIC) acquired in SCAN positive ionization mode (ESI+). Unlike what is normally done in targeted approaches, here deconvolution based on isotopic profile allows to use the EIC signal (extracted ion current), for defining features before identification. Based on this procedure, we decided not to include the UHPLC chromatogram as it provides limited information. Moreover, it's not feasible to present EIC chromatograms for every compound since this would involve thousands of peaks with varying treatments. We previously considered this justification adequate for excluding the figure, however, if you believe that the chromatograms would add value and clarity for the readers, we would be willing to include them in the supplementary material

  1. please indicate in Table S1 which polyphenolic compounds were identified for the first time (de novo), as claimed by the Authors (L483).

We are sorry that we did not express ourselves correctly with this word, so we have edited the text to avoid misunderstandings for readers and we have revised the entire text to correct any other expression that we cannot clearly demonstrate through our results.

Round 2

Reviewer 3 Report

The manuscript has been revised in accordance with the Reviewers' recommendations and is now suitable for publication in Antioxidants.